# FORWARD-LEARNED DISCRETE DIFFUSION: LEARNING HOW TO NOISE TO DENOISE FASTER

**Grigory Bartosh** [*]
University of Amsterdam
g.bartosh@uva.nl

**Teodora Pandeva, Sushrut Karmalkar & Javier Zazo**
Microsoft Research, Cambridge
{tpandeva,skarmalkar,javierzazo}@microsoft.com

## ABSTRACT

Discrete diffusion models are a powerful class of generative models with strong performance across many domains. For efficiency, however, discrete diffusion typically parameterizes the generative (reverse) process with factorized distributions, which makes it difficult for the model to learn the target process in a small number of steps and necessitates a long, computationally expensive sampling procedure. To reduce the gap between the target and model distributions and enable few-step generation, we propose Forward-Learned Discrete Diffusion (FLDD), which introduces discrete diffusion with a learnable forward (noising) process. Rather than fixing a Markovian forward chain, we adopt a non-Markovian formulation with learnable marginal and posterior distributions. This allows the generative process to remain factorized while matching the target defined by the noising process. We train all parameters end-to-end under the standard variational objective. Experiments on various benchmarks show that, for a given number of sampling steps, our approach produces a higher quality samples than conventional discrete diffusion models using the same reverse parameterization.

## 1 INTRODUCTION

In the last years, diffusion models have demonstrated strong performance across many continuous (Hoogeboom et al., 2024) and discrete (Lou et al.) domains . Recent work has shown that distillation approaches and advanced training techniques allow learning a few-step (Salimans et al., 2024), or sometimes even a single-step, generative (Xu et al., 2025) procedure in the continuous domain. However, discrete diffusion still requires a significant number of consecutive generative steps (often comparable to the data dimensionality), which makes inference computationally and time expensive.

The fundamental difference between continuous and discrete diffusion is the existence of a generative ordinary differential equation (ODE) in the continuous case. Such an ODE defines continuous trajectories that connect the noise and data distributions. Therefore, by learning the trajectories of an ODE, we can sample noise and directly map it to a data point. In discrete space, however, such continuous deterministic trajectories that would uniquely map noise to data do not exist, so these techniques are not applicable. At the same time, direct training of a few-step conventional diffusion model fails due to a mismatch between the complicated target distribution and the limited parameterization of the generative process. However, recent work on more flexible forward processes in diffusion models demonstrates that learning the noising process can improve generative performance in both continuous (Bartosh et al., 2024) and discrete Shi et al. settings.

Inspired by these results, we propose Forward-Learned Discrete Diffusion (FLDD) – a framework for discrete diffusion models with a highly flexible parameterization of the forward process. We show how to parameterize the forward process so that a training iteration remains computationally efficient, while the way we destroy information at each individual coordinate depends on the entire datapoint and not just the time step, as in conventional diffusion. Moreover, we show how to train this model efficiently in an end-to-end manner by minimizing the variational bound on the model's likelihood. The flexibility of FLDD enables better alignment of the two processes with a small number of steps, leading to improved generative performance.

---

[*]Work done while interning at Microsoft Research Cambridge.

We summarize our contributions as follows:

- We introduce Forward-Learned Discrete Diffusion (FLDD), generalizing discrete diffusion models and providing an efficient parameterization of the model.
- We provide an efficient, end-to-end, simulation-free training procedure.
- We demonstrate the robustness of our method on several discrete domains, including images, molecules, and natural language text.

## 2 BACKGROUND

### 2.1 AUTOREGRESSIVE MODELS

We aim to model a distribution over discrete sequences $q(\mathbf{x})$, where $\mathbf{x} \in \{1, \ldots, K\}^D$, $K$ is the number of discrete values, and $D$ is the length of the sequence $\mathbf{x}$. The standard approach for this is the autoregressive (AR) model, where each next token is predicted sequentially, conditioned on all previous ones. Formally, the density of the AR model can be written as:

$$p_\theta(\mathbf{x}) = \prod_{i=1}^{D} p_\theta(\mathbf{x}^i | \mathbf{x}^{<i}), \tag{1}$$

where $\mathbf{x}^i$ represent coordinate number $i$ of $\mathbf{x}$ and $\mathbf{x}^{<i}$ represents a prefix of first $i$ coordinates.

An important drawback of the AR approach is its slow sequential inference. To sample a sequence of $D$ elements, one must sample $D$ times in sequence from the conditional distribution $p_\theta(\mathbf{x}^i | \mathbf{x}^{<i})$.

### 2.2 DIFFUSION MODELS

In contrast, diffusion models (DMs) provide a fully parallel sampling procedure. A DM consists of two main components: the forward and reverse processes. The forward, or noising, process is a Markov process that takes a data point $\mathbf{x}$ and progressively corrupts it, producing a sequence of latent variables $\mathbf{z}_0, \mathbf{z}_1, \ldots, \mathbf{z}_T$:

$$q(\mathbf{z}_{0:T} | \mathbf{x}) = q(\mathbf{z}_0 | \mathbf{x}) \prod_{t=1}^{T} q(\mathbf{z}_t | \mathbf{z}_s), \tag{2}$$

where $s = t - 1$.

The reverse process is also a Markov chain. It starts from a sample $\mathbf{z}_T$ and proceeds backward in time, inverting the corruptions applied by the forward process:

$$p_\theta(\mathbf{z}_{0:T}, \mathbf{x}) = p(\mathbf{x} | \mathbf{z}_0) p(\mathbf{z}_T) \prod_{t=1}^{T} q(\mathbf{z}_s | \mathbf{z}_t), \tag{3}$$

where $p(\mathbf{x} | \mathbf{z}_0)$ is usually a simple, non-parametric distribution.

The training objective of diffusion models is a variational bound on the log-likelihood of the reverse process Ho et al. (2020):

$$-\log p_\theta(\mathbf{x}) \leq \underbrace{\mathbb{E}_{u(t)q(\mathbf{z}_t|\mathbf{x})} \left[ D_{\mathrm{KL}}\Big( q(\mathbf{z}_s|\mathbf{z}_t, \mathbf{x}) \| p_\theta(\mathbf{z}_s|\mathbf{z}_t) \Big) \right]}_{\mathcal{L}_{\mathrm{diff}}} + \tag{4}$$

$$\underbrace{\mathbb{E}_{q(\mathbf{z}_0|\mathbf{x})} \Big[ -\log p(\mathbf{x}|\mathbf{z}_0) \Big]}_{\mathcal{L}_{\mathrm{rec}}} + \underbrace{D_{\mathrm{KL}}\Big( q(\mathbf{z}_T|\mathbf{x}) | p(\mathbf{z}_T) \Big)}_{\mathcal{L}_{\mathrm{prior}}}, \tag{5}$$

where $u(t)$ is a uniform discrete distribution over the time steps $1, \ldots, T$. The three terms correspond to the reconstruction loss $\mathcal{L}_{\mathrm{rec}}$, the diffusion term $\mathcal{L}_{\mathrm{diff}}$, and the prior term $\mathcal{L}_{\mathrm{prior}}$.

Typically, the forward process is designed such that $\mathbf{z}_0 \approx \mathbf{x}$ and $q(\mathbf{z}_T|\mathbf{x}) \approx p(\mathbf{z}_T)$, where $p(\mathbf{z}_T)$ is a known prior. In practice, the reconstruction and prior terms are often negligible, and attention is usually focused on the diffusion term $\mathcal{L}_{\mathrm{diff}}$.

Although a single step in DMs, following $p_\theta(\mathbf{z}_s|\mathbf{z}_t)$, may have a computational cost comparable to sampling from $p_\theta(\mathbf{x})$ in the AR approach, DMs allow this process to be parallelized, with all coordinates updated simultaneously at each step. Thus, instead of $D$ sequential steps in AR models, DMs require $T$ sequential steps.

## 2.3 LIMITATIONS OF DIFFUSION MODELS

Despite this appealing formulation, in practice the number of steps $T$ required for DMs can be comparable to sequence length $D$, undermining the speed advantage. To see why DMs perform poorly when $T$ is small, we need to examine what they actually learn.

It can be shown that, for a given forward process, the target distribution for the reverse process is the marginalization of the posterior:

$$q(\mathbf{z}_s|\mathbf{z}_t) = \int q(\mathbf{x}|\mathbf{z}_t)q(\mathbf{z}_s|\mathbf{z}_t, \mathbf{x})d\mathbf{x} = \mathbb{E}_{q(\mathbf{x}|\mathbf{z}_t)}\Big[q(\mathbf{z}_s|\mathbf{z}_t, \mathbf{x})\Big]. \tag{6}$$

Thus, the forward process implicitly defines the target for the reverse process. If the reverse model $p_\theta(\mathbf{z}_s|\mathbf{z}_t)$ is not expressive enough to match the target distribution $q(\mathbf{z}_s|\mathbf{z}_t)$, it will fail to accurately approximate the data distribution $q(\mathbf{x})$.

At the same time, efficient sampling from $p_\theta(\mathbf{z}_s|\mathbf{z}_t)$ is essential for DM performance. The most straightforward way to enable fast, parallel sampling is to parameterize the reverse process as factorized:

$$p_\theta(\mathbf{z}_s|\mathbf{z}_t) = \prod_{i=1}^{D} p_\theta(\mathbf{z}_s^i|\mathbf{z}_t), \quad \text{where} \quad p_\theta(\mathbf{z}_s^i|\mathbf{z}_t) = \text{Cat}\Big(\mathbf{z}_s^i; \mathbf{v}_\theta^i(\mathbf{z}_t, t)\Big), \tag{7}$$

where $\mathbf{z}_s^i$ denotes the $i$-th coordinate of the discrete sequence $\mathbf{z}_s$.

This factorization allows all elements to be sampled in parallel, but it also greatly limits the model's flexibility. However, in general, the target distribution $q(\mathbf{z}_s|\mathbf{z}_t)$ in Equation (6) is not factorized. This difference is especially notable when the number of steps $T$ is small.

Importantly, the factorized parameterization in Equation (7) does not imply that each coordinate is treated independently. At step $s$, the distribution of each $\mathbf{z}_s^i$ depends on all coordinates of the previous state $\mathbf{z}_t = [\mathbf{z}_t^1, \dots, \mathbf{z}_t^D]$, not only on $\mathbf{z}_t^i$. Thus, even though sampling is performed through factorized distributions, the resulting distribution $p_\theta(\mathbf{x})$ can still be complex and non-factorized.

## 3 FORWARD-LEARNED DISCRETE DIFFUSION

To construct a discrete DM with only a few generative steps, we need to reduce the gap between the target distribution $q(\mathbf{z}_s|\mathbf{z}_t)$ (Equation (6)) and the model approximation $p_\theta(\mathbf{z}_s|\mathbf{z}_t)$ (Equation (7)). It is unclear how to make the reverse process more flexible without compromising efficiency. Instead, we propose to make the forward process more flexible. Then, the target distribution induced by the forward process (Equation (6)) can be adapted to the constraints of the generative process and enable few-step generation. To build intuition for the desired structure of the model, let us consider two motivational examples.

First, consider a mixture of two isotropic Gaussian distributions in $D$-dimensional space: $q(\mathbf{x}) = w\mathcal{N}(\mathbf{x}; \mu_1, I) + (1 - w)\mathcal{N}(\mathbf{x}; \mu_2, I)$. In the general case, we cannot sample from this distribution in a single step by drawing a sample from a factorized distribution, since the coordinates are correlated. However, we can do it in two steps: first sample the index of the Gaussian, and then sample from the chosen Gaussian. In this case, both samples come from factorized conditional distributions.

A less trivial example is a discrete random walk. Consider a process that starts at $\mathbf{x}^1 = 0$ and takes steps (either $+1$ or $-1$): $\mathbf{x}^{i+1} = \mathbf{x}^i \pm 1$. This process generates sequences of $D$ elements and can be naturally modeled in $D$ steps with an autoregressive approach. Interestingly, it can also be modeled in just two steps. First, sample $D - 1$ independent $\pm 1$ values, which define the individual steps. Then, by taking prefix sums, we can construct the full random-walk sequence.

These examples demonstrate that, in contrast to conventional discrete diffusion with uniform-noise injection or absorption (masking), some forward processes allow generation of complex data in a

small number of steps. The question is: what should this process be? Unfortunately, in the general case, it is unclear how the forward process should look. This question is especially challenging because the structure of the forward process must depend on the underlying data distribution. Therefore, in this work we propose learning it from data.

In this section, we present Forward-Learned Discrete Diffusion (FLDD) – a discrete diffusion framework that enables end-to-end training of both the forward and reverse processes, which leads to better likelihood estimation and allows a significant reduction in the number of generative steps. At the same time, our approach does not change the reverse process and, as a result, the generative procedure, so it introduces no inference overhead. FLDD also preserves important properties of conventional diffusion models, such as the variational objective and a simulation-free optimization procedure.

## 3.1 LEARNABLE FORWARD PROCESS

Since we want to modify the target (Equation (6)), we must change the forward process that induces it. Let us begin by generalizing the forward process.

The forward process is only required during training. From the objective in Equation (5), we observe that the forward process must satisfy two conditions: efficient sampling from the marginals $q(\mathbf{z}_t|\mathbf{x})$ and tractable posteriors $q(\mathbf{z}_s|\mathbf{z}_t, \mathbf{x})$ for evaluating the KL divergence. With this in mind, we can reformulate the forward process from its Markovian definition in Equation (2) to the following non-Markovian form:

$$q(\mathbf{z}_{0:T}|\mathbf{x}) = q(\mathbf{z}_T|\mathbf{x}) \prod_{t=1}^{T} q(\mathbf{z}_s|\mathbf{z}_t, \mathbf{x}). \tag{8}$$

This non-Markovian formulation enables a more flexible and efficient parameterization of the forward process. Specifically, we suggest making the forward-process distributions $q_\varphi(\mathbf{z}_t|\mathbf{x})$ and $q_\varphi(\mathbf{z}_s|\mathbf{z}_t, \mathbf{x})$ learnable. If the parameterization is sufficiently flexible, we can expect to find parameters $\varphi$ such that the resulting target distribution $q_\varphi(\mathbf{z}_s|\mathbf{z}_t)$ (Equation (6)) is factorized.

To learn the parameters $\varphi$, we optimize the same variational objective as in Equation (5). In this setup, just as the reverse process adapts to the forward process, the forward process also adapts to the reverse one. Since the generative process is factorized by design (Equation (7)), the forward process is encouraged to produce a factorized target distribution that the reverse process can effectively match.

## 3.2 PARAMETERIZATION OF THE FORWARD PROCESS

**Forward Marginals.** For the forward marginal distributions $q_\varphi(\mathbf{z}_t|\mathbf{x})$, we propose using the same form as the generative model (Equation (7)) and parameterizing them as factorized distributions:

$$q_\varphi(\mathbf{z}_t|\mathbf{x}) = \prod_{i=1}^{D} q_\varphi(\mathbf{z}_t^i|\mathbf{x}), \quad \text{where} \quad q_\varphi(\mathbf{z}_t^i|\mathbf{x}) = \text{Cat}\left(\mathbf{z}_t^i; \mathbf{u}_\varphi^i(\mathbf{x}, t)\right). \tag{9}$$

This parameterization allows us to sample $\mathbf{z}_t$ given $\mathbf{x}$ efficiently during training. However, unlike in conventional discrete diffusion models, each $\mathbf{z}_t^i$ may come from a complex distribution that depends on the entire data sample $\mathbf{x} = \mathbf{x}^1, \ldots, \mathbf{x}^D$, not just the $i$-th element $\mathbf{x}^i$.

In addition, we must enforce the boundary conditions $q_\varphi(\mathbf{z}_0|\mathbf{x}) = \delta(\mathbf{z}_0 - \mathbf{x})$ and $q_\varphi(\mathbf{z}_T|\mathbf{x}) = p(\mathbf{z}_T)$. Both properties can be ensured through an appropriate reparameterization of $\mathbf{u}_\varphi$.

**Forward Posteriors.** In addition to being tractable, the posterior distribution $q_\varphi(\mathbf{z}_s|\mathbf{z}_t, \mathbf{x})$ must be consistent with the marginals $q_\varphi(\mathbf{z}_t|\mathbf{x})$:

$$q_\varphi(\mathbf{z}_s|\mathbf{x}) = \int q_\varphi(\mathbf{z}_t|\mathbf{x}) q_\varphi(\mathbf{z}_s|\mathbf{z}_t, \mathbf{x}) d\mathbf{z}_t. \tag{10}$$

n other words, the posteriors $q_\varphi(\mathbf{z}_s|\mathbf{z}_t, \mathbf{x})$ must define a valid transport plan for the probability mass across consecutive time steps. We propose using a simple non-parametric technique, *Maximum Coupling*, for posterior parameterization.

---

**Algorithm 1** Training procedure with REINFORCE method.

---

**Require:** Dataset, $T$ – time-steps, $u_\varphi(\mathbf{x}, t)$ – forward parameterization, $\mathbf{v}_\theta(\mathbf{z}_t, t)$ – reverse parameterization

    **for** learning iterations **do**

        $\mathbf{x} \sim$ Dataset, $t \sim u(t)$, $s = t - 1$     $\triangleright$ Sampling data point $\mathbf{x}$ and two consecutive time steps

        $\mathbf{u}_s = u_\varphi(\mathbf{x}, s)$, $\mathbf{u}_t = u_\varphi(\mathbf{x}, t)$     $\triangleright$ Parameters of $q_\varphi(\mathbf{z}_s|\mathbf{x})$ and $q_\varphi(\mathbf{z}_t|\mathbf{x})$ (Equation (9))

        $\mathbf{z}_t \sim \mathrm{Cat}(z_t; \mathbf{u}_t)$     $\triangleright$ Sampling $\mathbf{z}_t$ from categorical distribution with parameters $\mathbf{u}_t$

        Construct $\mathbf{u}_{s|t}$ by $\mathbf{u}_s$, $\mathbf{u}_t$ and $\mathbf{x}$     $\triangleright$ Calculating parameters of $q_\varphi(\mathbf{z}_s|\mathbf{z}_t, \mathbf{x})$ (Section 3.2)

        $\mathbf{v}_{s|t} = \mathbf{v}_\theta(\mathbf{z}_t, t)$     $\triangleright$ Parameters of the reverse process $p_\theta(\mathbf{z}_s|\mathbf{z}_t)$ (Equation (7))

        $\mathcal{L} = \sum_{i=1}^{D} \mathbf{u}_{s|t}^i \log \frac{\mathbf{u}_{s|t}^i}{\mathbf{v}_{s|t}^i}$     $\triangleright$ Objective function is KL divergence (Equation (5))

        Gradient step on $\theta$ and $\varphi$ w.r.t. $\frac{\langle \mathbf{u}_t, \mathbf{z}_t \rangle}{\lfloor \langle \mathbf{u}_t, \mathbf{z}_t \rangle \rfloor_{\mathrm{sg}}} \mathcal{L}$     $\triangleright$ REINFORCE functional (Section 3.3)

    **end for**

---

Suppose we want to construct a transport plan between two 1-dimensional categorical distributions with parameters $\mathbf{u}_s, \mathbf{u}_t \in \Delta^{K-1}$. Intuitively, when moving probability mass from $\mathbf{u}_t$ to $\mathbf{u}_s$, Maximum Coupling attempts to minimize the amount of mass that must be moved. Specifically, for $\mathbf{z}_t = k$, if $\mathbf{u}_s^k \geq \mathbf{u}_t^k$, we keep $\mathbf{z}_s = \mathbf{z}_t$. Otherwise, if $\mathbf{u}_s^k < \mathbf{u}_t^k$, we redistribute the excess mass $\mathbf{u}_t^k - \mathbf{u}_s^k$ from bin $k$ to bins with a deficit.

Formally, we can write it as:

$$q(\mathbf{z}_s|\mathbf{z}_t) = \mathrm{Cat}(\mathbf{z}_s; \mathbf{u}_{s|t}), \quad \text{where} \quad \mathbf{u}_{s|t}^j = \begin{cases} \frac{\min(\mathbf{u}_s^k, \mathbf{u}_t^k)}{\mathbf{u}_t^k}, & z_t = k = j, \\ \frac{\min(0, \mathbf{u}_s^k - \mathbf{u}_t^k)}{\mathbf{u}_s^k} \mathbf{m}_{s|t}^j, & z_t = k \neq j. \end{cases}$$

$$\text{and} \quad \mathbf{m}_{s|t} = \frac{\min(0, \mathbf{u}_s - \mathbf{u}_t)}{\|\min(0, \mathbf{u}_s - \mathbf{u}_t)\|}. \tag{11}$$

Here, $\mathbf{m}_{s|t}$ represents the parameters of a categorical distribution corresponding to the deficit of probability mass at timestep $s$ compared to timestep $t$. We redistribute the extra probability mass from $\mathbf{u}_t$ to $\mathbf{u}_s$ according to $\mathbf{m}_{s|t}$. Importantly, the computation of the posterior parameters requires only simple vector operations, so it can be performed efficiently.

To construct the full $D$-dimensional distribution $q_\varphi(\mathbf{z}_s|\mathbf{z}_t, \mathbf{x})$, we apply the Maximum Coupling procedure independently to each coordinate, using the parameters $\mathbf{u}_s = \mathbf{u}_\varphi(\mathbf{x}, s)$ and $\mathbf{u}_t = \mathbf{u}_\varphi(\mathbf{x}, t)$ (Equation (9)).

We emphasize that, with this parameterization, the distributions of forward trajectories for each coordinate are conditionally independent given the data point $\mathbf{x}$. However, in contrast to conventional discrete diffusion, the trajectory of each coordinate $\mathbf{z}_t^i$ may have a complex, non-linear dependence on the entire datapoint $\mathbf{x}$, not just on $\mathbf{x}^i$. Moreover, marginal distributions can still be complex, and the unconditional distribution of forward trajectories remains expressive.

The parameterization of the forward process described here is not unique. For instance, we may trade off sampling efficiency of the marginals for additional flexibility by making them partially autoregressive. For the posterior, we could use element-wise optimal transport with respect to a chosen metric or introduce dependencies between coordinates or additional learnable parameters. We believe there may be better ways to parameterize the forward process, but we leave this question for future research.

## 3.3 OPTIMIZATION

As mentioned in Section 3.1, we train the model by minimizing the same variational objective as in conventional diffusion (Equation (5)) with respect to the parameters $\theta$ and $\varphi$. While the objective itself is fully tractable, its gradients with respect to $\varphi$ are difficult to compute. Let us take a closer look at the gradients of the diffusion term:

$$\nabla_\varphi \mathcal{L}_{\mathrm{diff}} = \nabla_\varphi \mathbb{E}_{u(t) q_\varphi(\mathbf{z}_t|\mathbf{x})} \left[ \mathrm{D}_{\mathrm{KL}}\Big( q_\varphi(\mathbf{z}_s|\mathbf{z}_t, \mathbf{x}) \| p_\theta(\mathbf{z}_s|\mathbf{z}_t) \Big) \right]. \tag{12}$$

---

**Algorithm 2** Sampling procedure.

---

**Require:** $T$ – time-steps, $\mathbf{v}_\theta(\mathbf{z}_t, t)$ – reverse parameterisation

$\quad \mathbf{z}_T \sim \text{Cat}(\mathbf{z}_T; \mathbf{v}_T)$ $\qquad\qquad\qquad\qquad$ ▷ Sampling $\mathbf{z}_T$ from prior distribution $p(\mathbf{z}_T)$

$\quad$ **for** $t \in T, \dots, 1$ **do**

$\qquad \mathbf{v}_{s|t} = \mathbf{v}_\theta(\mathbf{z}_t, t)$ $\qquad\qquad$ ▷ Parameters of the reverse process $p_\theta(\mathbf{z}_s|\mathbf{z}_t)$ (Equation (7))

$\qquad \mathbf{z}_s \sim \text{Cat}(\mathbf{z}_s; \mathbf{v}_{s|t})$ $\qquad$ ▷ Sampling $\mathbf{z}_t$ from categorical distribution with parameters $\mathbf{v}_{s|t}$

$\quad$ **end for**

---

The expression inside the expectation is fully differentiable. However, the distribution $q_\varphi(\mathbf{z}_t|\mathbf{x})$ and the samples $\mathbf{z}_t$ are discrete, which prevents us from using the reparameterization trick to efficiently estimate gradients.

**Unbiased Optimization.** The standard approach for computing gradients of this form is the REIN-FORCE method Williams (1992). We can rewrite the gradients of the diffusion loss as follows:

$$\nabla_\varphi \mathcal{L}_{\text{diff}} = \mathbb{E}_{u(t)q_\varphi(\mathbf{z}_t|\mathbf{x})} \left[ \nabla_\varphi \left( \frac{q_\varphi(\mathbf{z}_t|\mathbf{x})}{\lfloor q_\varphi(\mathbf{z}_t|\mathbf{x}) \rfloor_{\text{sg}}} D_{\text{KL}} \Big( q_\varphi(\mathbf{z}_s|\mathbf{z}_t, \mathbf{x}) \| p_\theta(\mathbf{z}_s|\mathbf{z}_t) \Big) \right) \right], \qquad (13)$$

where $\lfloor \cdot \rfloor_{\text{sg}}$ is a stop-grad operation.

In this way, we can use the Monte Carlo method to build an unbiased gradient estimator and train the model end-to-end. Importantly, as in conventional diffusion, we optimize a variational bound on the model's likelihood. We summarize the training algorithm in Algorithm 1.

**Relaxed Warm-Up.** Unfortunately, REINFORCE is known to produce high-variance gradients, so training a model from scratch with REINFORCE alone often leads to instability. To obtain a better initialization, we adopt a different strategy.

We begin training with a continuous relaxation of the categorical distribution $q_\varphi(\mathbf{z}_t|\mathbf{x})$ using the Concrete distribution Jang et al. (2016); Maddison et al. (2016). Specifically, instead of drawing hard samples $\mathbf{z}_t$ according to Equation (9), we use the same parameters $\mathbf{u}_\varphi(\mathbf{x}, t)$ to define a Concrete distribution $\bar{q}_{\tau,\varphi}(\bar{\mathbf{z}}_t|\mathbf{x}) \approx q_\varphi(\mathbf{z}_t|\mathbf{x})$ with temperature $\tau$. To construct a posterior for a relaxed sample $\bar{\mathbf{z}}_t$, we combine the posteriors for discrete samples $\mathbf{z}_t$ according to the components of $\bar{\mathbf{z}}_t$:

$$\bar{q}_{\tau,\varphi}(\mathbf{z}_s^i|\bar{\mathbf{z}}_t^i, \mathbf{x}) = \sum_{k=1}^{K} \bar{\mathbf{z}}_t^{i,k} q_\varphi(\mathbf{z}_s^i|\mathbf{z}_t^i = k, \mathbf{x}), \qquad (14)$$

where $\bar{\mathbf{z}}_t^{i,k}$ denotes the component corresponding to discrete value $k$ of coordinate $i$ in the relaxed sample $\bar{\mathbf{z}}_t$. Thanks to the simple structure of the posterior distribution (Section 3.2), this weighted average does not add computational overhead.

Relaxed samples allow us to use the reparameterization trick for gradient estimation. Training starts with a temperature $\tau = 1$, which we exponentially decrease to $\tau = 10^{-3}$ over $10^4$ to $10^5$ optimization steps. After this warm-up phase, we switch to training the model with the REINFORCE approach described above.

### 3.4 GENERATIVE PROCESS

We use the standard parameterization of the generative process as discussed in Section 2.2 (Equation (7)). Importantly, the sampling procedure does not require the forward process, so we do not add any computational overhead for inference. We summarize the generative procedure in Algorithm 2.

It is important to note that another common parameterization, where the model samples an approximate data point $\hat{\mathbf{x}}$ and then resamples an intermediate step $q_\varphi(\mathbf{z}_s|\mathbf{z}_t, \hat{\mathbf{x}})$, is not applicable in our formulation. Although it may be possible to make $q_\varphi(\mathbf{z}_s|\mathbf{z}_t)$ (Equation (6)) factorized, the distribution $q_\varphi(\mathbf{x}|\mathbf{z}_t)$ will generally not be factorizable. As a result, the reverse process cannot learn it accurately.

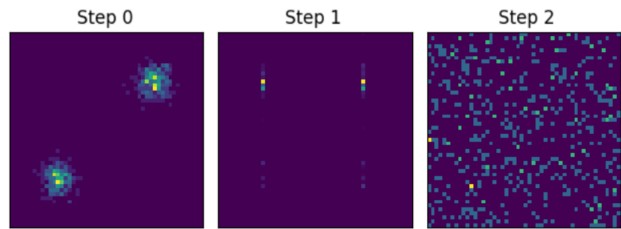

Figure 1: Training data distribution and learned dynamics for a mixture of two Gaussians.

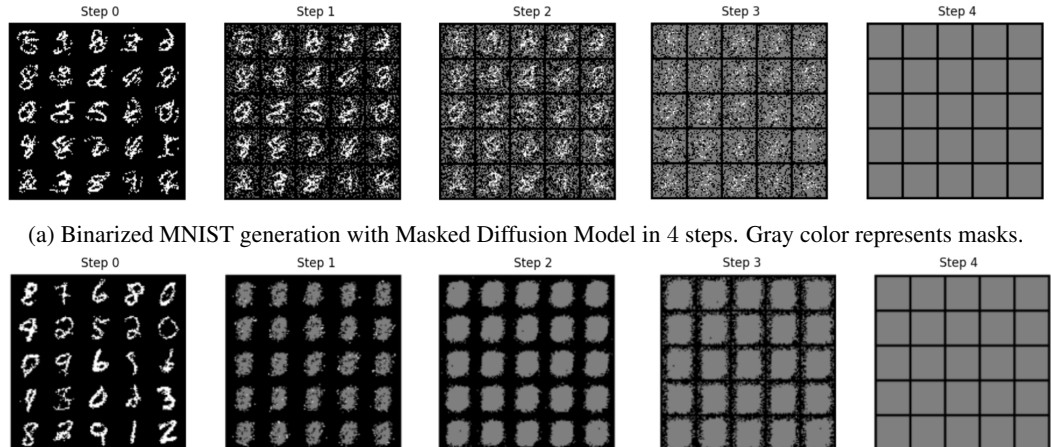

(a) Binarized MNIST generation with Masked Diffusion Model in 4 steps. Gray color represents masks.

(b) Binarized MNIST generation with learned forward process in 4 steps.

Figure 2: Learned dynamics for Binarized MNIST dataset. Generative process starts from prior distribution $p(z_T)$ at time steps $t = T$ and goes backwards in time.

## 4 EXPERIMENTS

In this section, we provide experiments on both synthetic and real-world data. We present FLDD as a general framework for reducing the number of sampling steps. Therefore, the goal of this section is to demonstrate that FLDD indeed improves sampling efficiency across different settings. Our objective is not to outperform all other approaches in every domain. Instead, FLDD is compatible with existing extensions. We also assume there is potential room for optimizations and better design choices in these domains.

In all our experiments, we use the same neural network architectures and hyperparameters to parameterize both the generative process $p_\theta(\mathbf{z}_s|\mathbf{z}_t)$ and the forward marginals $q_\varphi(\mathbf{x}|\mathbf{z}_t)$. We emphasize that, while this doubles the number of learnable parameters, it does not affect the capacity of the generative process. Additional details on parameterization, training, and evaluation are provided in Section A.

### 4.1 TOY DATA

First, we consider a toy setting with $\mathbf{x} \in \{1, \ldots, 50\}^2$. We train a diffusion model to generate samples in just two steps. In Figure 1, we show results for a mixture of two Gaussian distributions. As we discussed in Section 3, each individual Gaussian can be generated in a single step, since the coordinates are independent. However, generating the mixture requires at least two steps. In this case, the model learns first to generate an intermediate factorized structure and then to produce the final data.

Table 1: Results on ROCStories dataset.

| | MAUVE ↑ | PPL ↓ | Div ↑ |
|---|---|---|---|
| GPT2 (Radford et al., 2019) | 0.789 | 20.5 | 0.252 |
| GPT Neo (Black et al., 2021) | 0.720 | 19.9 | 0.258 |
| AR-Diffusion (Wu et al., 2023) | 0.066 | 41.8 | 0.101 |
| DiffuSeq (Gong et al., 2022) | 0.086 | 50.5 | 0.124 |
| SeqDiffuSeq (Yuan et al., 2022) | 0.103 | 29.3 | 0.137 |
| TESS (Mahabadi et al., 2023) | 0.061 | 22.4 | 0.163 |
| SEDD (Lou et al., 2023) | 0.598 | 70.8 | 0.336 |
| LD4LG (Lovelace et al., 2023) | 0.716 | 30.6 | 0.331 |
| COSMOS (Meshchaninov et al., 2025) | 0.940 | 26.3 | 0.346 |
| FLDD, $T = 100$ | 0.538 | 55.2 | 0.280 |
| FLDD, $T = 10$ | 0.511 | 60.5 | 0.285 |

Table 2: Molecular generation results.

| | QM9 | | | ZINC250k | | |
|---|---|---|---|---|---|---|
| | Valid ↑ | Unique ↑ | FCD ↓ | Valid ↑ | Unique ↑ | FCD ↓ |
| MoFlow (Zang & Wang, 2020) | 91.36 | 98.65 | 4.467 | 63.11 | 99.99 | 20.931 |
| EDP-GNN (Niu et al., 2020) | 47.52 | 99.25 | 2.680 | 82.97 | 99.79 | 16.737 |
| GraphEBM (Liu et al., 2021) | 8.22 | 97.90 | 6.143 | 5.29 | 98.79 | 35.471 |
| GDSS (Jo et al., 2022) | 95.72 | 98.46 | 2.900 | 97.01 | 99.64 | 14.656 |
| Digress Vignac et al. (2022) | 99.00 | 96.20 | - | - | - | - |
| Flow Matching (Lipman et al., 2023) | 94.10 | 98.20 | 5.155 | 94.01 | 96.68 | 18.764 |
| Dirichlet FM (Stark et al., 2024) | 99.10 | 98.15 | 0.888 | 97.52 | 99.20 | 14.222 |
| CatFlow (Eijkelboom et al., 2024) | 99.81 | 99.95 | 0.441 | 99.21 | 100.00 | 13.211 |
| LO-ARM (Wang et al., 2025) | 99.85 | 98.85 | 0.240 | 96.70 | 100.00 | 3.229 |
| FLDD, $T = 100$ | 99.67 | 98.93 | 0.328 | 97.79 | 100.00 | 8.487 |
| FLDD, $T = 10$ | 99.08 | 99.95 | 0.385 | 96.77 | 100.00 | 10.414 |

## 4.2 REAL WORLD DATA

We evaluate FLDD on the text dataset ROCStories (Mostafazadeh et al., 2016) and two molecular generation benchmarks: QM9 (Ramakrishnan et al., 2014) and ZINC250k (Irwin et al., 2012). The results are summarized in Table 1 and Table 1. For each problem, we evaluate FLDD in two settings: first, with $T = 100$, which is comparable to what other models use; and second, with a significantly reduced $T = 10$ steps.

In all cases, FLDD demonstrates results comparable to other baselines with $T = 100$. Some baselines may outperform FLDD, but we attribute this to suboptimal parameterization and hyperparameters that we do not tune. As we discuss in Section 3.4, it is important for FLDD to parameterize the distribution $q_\varphi(\mathbf{z}_s|\mathbf{z}_t)$ directly, which is known to be suboptimal in conventional diffusion. We believe there are ways to reparameterize the generative process efficiently while preserving the same properties. However, we leave this for future research.

At the same time, FLDD shows only a slight drop in performance with $T = 10$ steps, whereas other diffusion models do not generate realistic-looking data with such a small number of steps. This demonstrates that FLDD allows a significantly better trade-off between sample quality and inference speed.

## 4.3 LEARNABLE MASK DIFFUSION MODELS

A popular variation of discrete diffusion is mask discrete models (MDM), where the forward process randomly replaces tokens from the sequence $\mathbf{x}$ with masks, and the reverse process replaces mask tokens with tokens from the vocabulary. For MDM, it is clear that if there is correlation between coordinates and the number of generative steps $T$ is significantly smaller than the sequence length $D$, the generative model will fail, since it will have to unmask tokens independently without taking

correlations into account. In the general case, this problem is unavoidable in conventional MDM. However, with FLDD, we can improve performance by learning to unmask at each step only tokens that are less correlated (conditionally on current observations).

To demonstrate how this works, we restrict the forward process to masking. Specifically, $q_\varphi(\mathbf{z}_t|\mathbf{x})$ defines only the probability of masking each token conditionally on the full data point $\mathbf{x}$. Then, we expect the model to learn masking schedules such that the generative process corresponds to unmasking less correlated tokens.

We conduct an experiment on the MNIST dataset Salakhutdinov & Hinton (2009). Figure 2 shows images generated by a conventional MDM and by FLDD with learnable masking schedules. Since an MDM learns to unmask pixels uniformly, it struggles to generate natural-looking images. In contrast, our model learns more convenient intermediate distributions and produces more realistic samples. We emphasize that the generative processes in both models share the same parameterization.

## 5 RELATED WORK

Diffusion models Ho et al. (2020) have recently been adapted to discrete spaces for language, graphs, and molecules. Text diffusion includes sequence-to-sequence formulations Gong et al. (2022); Yuan et al. (2022), hybrid or autoregressively-conditioned variants Wu et al. (2023), simplex- or ratio-based objectives Mahabadi et al. (2023); Lou et al. (2023), and masked/absorbing-state processes with improved schedules and theory Shi et al.. For molecular and graph data, flow-/score-based and diffusion-style approaches continue to advance sampling quality and validity Lipman et al. (2023); Stark et al. (2024); Eijkelboom et al. (2024); Jo et al. (2022); Vignac et al. (2022); Wang et al. (2025). Reducing the number of reverse steps has been pursued via distillation and consistency-style accelerations in the continuous domain Salimans et al. (2024); Xu et al. (2025), but discrete models often retain long sampling chains due to factorized reverse parameterizations. Our work targets this limitation by learning a forward process that induces a reverse target distribution compatible with the same efficient, factorized generative family, thereby enabling few-step sampling without changing the sampler itself.

Most diffusion methods fix the forward (noising) dynamics, which implicitly defines the target that the reverse model must match. Recent work in the *continuous* setting shows that learning the forward process can tighten likelihood bounds and improve generation Bartosh et al. (2024); Shi et al.. We build on this thread but in the *discrete* regime: we introduce a non-Markovian yet tractable parameterization of the forward marginals and posteriors, and train it end-to-end together with the standard factorized reverse model. This design preserves the usual variational training objective and the parallel, stepwise sampler, while adapting the training target to better match the model's inductive biases.

## 6 CONCLUSION

We presented Forward-Learned Discrete Diffusion (FLDD), a framework that learns the forward (noising) process—via a non-Markovian factorization of forward marginals and tractable posteriors—while keeping the standard factorized reverse sampler unchanged and preserving the usual variational training objective. This alignment between a learned target and the fixed reverse family enables few-step generation without inference overhead.

Empirically, FLDD matches strong baselines when $T{=}100$ and maintains competitive quality even at $T{=}10$ across text (ROCStories) and molecule benchmarks (QM9, ZINC250k), and improves masked-diffusion generation on images by learning data-aware masking schedules—all with the same reverse parameterization. This yields a better quality–latency trade-off than conventional discrete diffusion.

Limitations and future work include exploring richer forward parameterizations, reducing reliance on REINFORCE via lower-variance estimators. We also note the extra parameters of the forward network (though not affecting reverse capacity) and leave broader design optimizations to future research.

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

## A    IMPLEMENTATION DETAILS

In all our experiments, we use identical neural network architectures and hyperparameters for both the generative process $p_\theta(\mathbf{z}_s|\mathbf{z}_t)$ and the forward marginals $q_\varphi(\mathbf{x}|\mathbf{z}_t)$. First, we warm up the model for $10^5$ steps as discussed in Section 3.3, then optimize with REINFORCE for $100k$ iterations. We use the AdamW optimizer with a learning rate of $2 \cdot 10^{-4}$.

For evaluation on the molecular datasets QM9 (Ramakrishnan et al., 2014) and ZINC250k (Irwin et al., 2012), we follow the standard setup in Shi et al. (2019); Luo et al. (2021); Vignac et al. (2022); Jo et al. (2022). We use the same hyperparameters for model parameterization as in Eijkelboom et al. (2024).

For the experiment on the ROCStories (Mostafazadeh et al., 2016) dataset, we use LLaMA 2 Touvron et al. (2023) with 6 layers and 6 heads and an embedding size of $512$ to parameterize the model, and a pretrained GPT-2 for PPL calculation.

