# OpenReview forum: "Forward-Learned Discrete Diffusion: Learning how to noise to denoise faster"
_ICLR.cc/2026/Conference — ICLR 2026 Poster_

### Official Review · Reviewer_wcR4 · 2025-10-29

**Soundness:** 2
**Presentation:** 2
**Contribution:** 4
**Rating:** 8
**Confidence:** 4

**Summary:**

The paper learns the forward noising process in a non-Markovian way so that the training targets match what a factorized reverse model can learn well in just a few steps. It keeps the reverse sampler unchanged. Via learned forward, the paper achieves much better quality–inference latency trade-offs. Across text (ROCStories) and molecular (QM9, ZINC250k) data, the paper with only $T=10$ reverse steps delivers quality close to its $T=100$ counterpart—unlike conventional discrete diffusion.

**Strengths:**

FLDD’s main idea is both theoretically sound and practically useful: learn the forward noising process in a manner so that the task for the reverse process becomes easier. In addition, the algorithm uses the standard variational objective and does not change the reverse sampler. The posterior is implemented with a cheap Maximum Coupling transport between consecutive marginals, which keeps training tractable while letting each coordinate depend on the whole input. Empirically, the paper shows few-step generation benefits illustrating a better quality–latency trade-off than conventional discrete diffusion.

**Weaknesses:**

FLDD adds a learned forward network, increasing training compute relative to standard discrete diffusion. The paper does not report quality vs total training and inference budget (e.g., GPU-hours or FLOPs), leaving unclear whether the few-step gains persist under fixed total compute budget.

**Questions:**

The paper does not compare REINFORCE against other standard gradient estimators for discrete latents. It does not discuss control variates such as Rao–Blackwellization tricks. These would reveal which knobs actually make training computationally stable/cheap. Could you report some of these results?

---

> ### Author Response · Authors · 2025-11-19
>
> We are pleased that the reviewer finds our work theoretically sound and practically useful. Below, we address the questions raised in the review.
>
> **Weaknesses:**
>
> A single training iteration of FLDD takes approximately twice as long as that of conventional diffusion models that do not parameterise the forward process. A higher computational budget per training iteration is an inevitable consequence of a more flexible forward process. However, we would like to emphasise that the exact factor is a consequence of our design choices. For simplicity of the experimental setup, we use neural networks of the same size to parameterise both the forward and reverse processes. We believe that the parameterisation of the forward process can be made significantly lighter than that of the reverse one, but we leave this hyperparameter optimisation for future work.
>
> We also note that the forward process is not involved in the generative procedure at all. Thus, while FLDD reduces the number of steps, it does not introduce any additional computational overhead at sampling time.
>
> In the camera-ready version, if accepted, we will provide a detailed report of the computational budget.
>
> **Questions:**
>
> We experimented with several control variates, such as straight-through Gumbel, Rao–Blackwellization, leave-one-out, REBAR, and RELAX. We found that the simple combination (described in the paper) of a warm-up phase with a Gumbel distribution followed by REINFORCE converges faster and more stably across different domains. Replacing the Gumbel distribution during the warm-up stage slows convergence. In the final stage, biased methods hurt performance, and more complicated unbiased methods (such as REBAR) do not help.
>
> We will provide a detailed discussion of these control variate methods and their comparison in the appendix of the camera-ready version, if accepted.

---

### Official Review · Reviewer_vLZq · 2025-10-30

**Soundness:** 3
**Presentation:** 3
**Contribution:** 3
**Rating:** 6
**Confidence:** 3

**Summary:**

This paper proposes forward-learned discrete diffusion (FLDD). It’s a discrete diffusion with a learnable forward process. The idea is to keep the generative process factorized while matching the target, so it can work well for small numbers of diffusion steps. Experiments show that the performance of FLDD drops a little when the number of diffusion steps goes down from 100 to 10.

**Strengths:**

- Novel approach for training discrete diffusion models

**Weaknesses:**

- FLDD with T=100 doesn’t outperform some previous work. The authors argue that FLDD performs better for small Ts. However, there are methods to accelerate sampling of discrete diffusion models, e.g. distillation. For small-T experiments, FLDD needs to be compared with methods focusing on accelerated sampling

**Questions:**

- What's the batch size used in training and how does it affect the performance?
- What are the additional costs of training, in terms of memory (chips) and duration (time)?

---

> ### Author Response · Authors · 2025-11-19
>
> We are delighted that the reviewer finds our approach novel. Below, we address the questions raised in the review.
>
> **Weaknesses:**
>
> The main goal of our work is not to surpass all specialised text models when using many sampling steps, but rather to demonstrate the robust efficiency of FLDD with a small number of steps across different data distributions. In Section 4, we show that FLDD achieves performance comparable to the baselines when a large number of steps is allowed, and that its performance deteriorates only modestly when using only a few steps—precisely in regimes where the baselines are not able to generate meaningful samples.
>
> Distillation is indeed a potential way to train a few-step discrete generator. However, diffusion-based distillation methods usually rely on existing continuous dynamics (in continuous space) and differentiable mappings, none of which are available in the discrete case. Moreover, diffusion distillation itself significantly complicates training, increases its computational cost, and is known to behave unstably. All of this makes it particularly difficult to learn a few-step generative procedure for discrete data, and this is precisely the challenge that FLDD addresses.
>
> We therefore view distillation of discrete diffusion as an interesting but separate line of work, which is moreover potentially compatible with FLDD. In the camera-ready version, if accepted, we will expand Sec. 5 to better position FLDD with respect to distillation-based approaches.
>
> **Questions:**
>
> 1. In all our experiments, we use a batch size of 1024. We have not heavily experimented with this hyperparameter.
> 2. A single training iteration of FLDD takes approximately twice as long as that of conventional diffusion models that do not parameterise the forward process. A higher computational budget per training iteration is an inevitable consequence of a more flexible forward process. However, we would like to emphasise that the exact factor is a consequence of our design choices. For simplicity of the experimental setup, we use neural networks of the same size to parameterise both the forward and reverse processes. We believe that the parameterisation of the forward process can be made significantly lighter than that of the reverse one, but we leave this hyperparameter optimisation for future work.
>
>     We also note that the forward process is not involved in the generative procedure at all. Thus, while FLDD reduces the number of steps, it does not introduce any additional computational overhead at sampling time.
>
>
> We hope that these clarifications address your concerns and will strengthen your support for the acceptance of our paper!

---

### Official Review · Reviewer_ATG6 · 2025-10-31

**Soundness:** 2
**Presentation:** 2
**Contribution:** 2
**Rating:** 4
**Confidence:** 3

**Summary:**

This paper introduces Forward-Learned Discrete Diffusion (FLDD), a new framework for discrete diffusion models aimed at improving sampling efficiency and enabling few-step generation. The core idea is to replace the standard fixed, Markovian noising process with a learnable, non-Markovian forward process. By co-training this flexible forward process alongside the reverse (denoising) process, the authors argue that the model can learn an easier-to-invert corruption path. This, in turn, allows a simple, factorized reverse sampler to generate high-quality samples in a significantly reduced number of steps. The method is trained end-to-end using the standard variational objective and is evaluated on synthetic data, binarized MNIST, text generation (ROCStories), and molecular generation (QM9, ZINC250k).

**Strengths:**

1. The paper does a good job of grounding the proposed method in the established variational inference framework for diffusion models. The formulation for end-to-end training of both the forward and reverse processes while preserving the standard variational objective is principled and clearly presented.
2. The paper is generally well-written and easy to follow.

**Weaknesses:**

1. The primary qualitative results are presented on a 2D Gaussian mixture and binarized MNIST. These are considered solved or overly simplistic problems in the current deep generative modeling landscape. Demonstrating success on these toy tasks provides very little signal about the method's effectiveness on complex, high-dimensional discrete data that is of interest to the community.
2. A significant weakness of this paper is its failure to demonstrate a clear performance advantage over existing, strong baselines. The paper's own results on the ROCStories dataset (Table 1) show that SEDD (Lou et al., 2023), a recent and relevant discrete diffusion model, achieves a higher MAUVE score (0.598 vs. 0.538) and a better (lower) PPL than the proposed FLDD, even when FLDD uses a large number of steps (T=100). This directly undermines the paper's central claims regarding improved sampling efficiency and achieving a better quality-latency trade-off. Given that a method from 2023 already shows superior performance on the same benchmark, the practical contribution of the proposed, more complex FLDD framework is called into question.
3. The paper fails to cite, discuss, or compare against the seminal work: LLaDA[1], a landmark paper that demonstrated how to successfully apply diffusion models to high-quality text generation and is arguably the most famous work in this specific area.

[1] Nie, Shen, Fengqi Zhu, Zebin You, Xiaolu Zhang, Jingyang Ou, Jun Hu, Jun Zhou, Yankai Lin, Ji-Rong Wen, and Chongxuan Li. "Large language diffusion models." arXiv preprint arXiv:2502.09992 (2025).

**Questions:**

N/A

---

> ### Author Response · Authors · 2025-11-19
>
> We thank the reviewers for their valuable feedback, which helps us improve the paper, and for the positive remarks on the formulation and writing. Below, we address the questions and comments raised in the reviews.
>
> **Weaknesses:**
>
> 1. We agree that 2D mixtures and binarized MNIST alone would not justify a new framework. That’s why in our work they serve only as *didactic* examples: the 2D mixture (Sec. 4.1, Fig. 1) visualizes how FLDD learns factorized intermediate marginals that still support 2-step generation, and binarized MNIST (Sec. 4.3, Fig. 2) illustrates how a learned masking schedule improves the few-step unmasking generative process.
>
>     The main evidence for FLDD on nontrivial, high-dimensional discrete data comes from text and molecular generation (Sec. 4.2, Tables 1–2). ROCStories is a standard benchmark for discrete diffusion text generation, and QM9/ZINC250k are widely used for molecule generation, where we compare against strong diffusion/flow baselines. In all cases, FLDD achieves generative performance comparable to baseline methods under a similar computational budget, while losing only a small amount of performance when the number of steps is significantly reduced (regimes in which other models fail to produce any meaningful samples at all).
>
>     We will make this emphasis clearer in Sec. 4 and will add more qualitative examples for ROCStories and QM9/ZINC250k in the appendix to better highlight performance on realistic data.
>
> 2. First, we would like to note that SEDD (Lou et al., 2023) doesn’t outperform FLDD. They have comparable MAUVE, while FLDD demonstrates better PPL for both $T=100$ and $T=10$.
>
>     Second, we would like to emphasize that the main point of our work is not to surpass all specialized text models at a large number of steps, but to demonstrate robust efficiency of FLDD with a small number of steps across different data distributions. In Section 4 we demonstrate that FLDD has performance comparable to the baselines with a large number of steps and deteriorates only modestly when using few steps, precisely in regimes where baselines are not able to generate meaningful samples. We demonstrate this consistent behavior across an image dataset, a text dataset, and two molecular datasets. We believe these results clearly support our claim of improved performance of FLDD as a few-step discrete generative framework.
>
>     We believe that with better parameterization, hyperparameter tuning, and the use of domain-specific techniques (e.g. extra losses, classifier-free guidance, self-conditioning, etc.), FLDD can demonstrate even better performance on applied tasks, but we leave this for future research.
>
> 3. We thank the reviewer for pointing out LLaDA (Nie et al., 2025) and agree that it is an important reference for large-scale diffusion language models. We will add it to the related work and clarify the relationship.
>
>     However, we would like to clarify that, first, LLaDA is an LLM with a size and training budget that are orders of magnitude higher than FLDD’s. Second, we present FLDD as a few-step discrete diffusion framework. While text is an important discrete domain, it is not unique. It serves as one of several examples rather than the sole focus of our work.
>
>     We would also like to note that FLDD is not a competitor, but is compatible with MDMs like LLaDA. Beyond applying FLDD framework directly to train a few-step LLaDA, one could take a pre-trained LLaDA model and use FLDD to learn just a masking/unmasking schedule to sample from LLaDA faster. However, this is beyond the scope of this work.
>
>
> We hope these clarifications, especially the corrected interpretation of the experiments, the emphasis on the few-step performance of FLDD, and the added discussion of LLaDA, address your concerns and will strengthen your support for acceptance!

---

### Official Review · Reviewer_mvjP · 2025-11-05

**Soundness:** 3
**Presentation:** 3
**Contribution:** 3
**Rating:** 6
**Confidence:** 3

**Summary:**

This paper introduces Forward-Learned Discrete Diffusion, a method designed to accelerate the reverse process while supporting end-to-end training.

Unlike previous works, FLDD adopts a non-Markovian formulation with learnable marginal and posterior distributions, which allows the generative process to remain factorized while matching the target distribution defined by the noising process. The proposed method enables a better alignment between the forward and reverse processes even with a small number of steps, thereby improving its generation performances.

**Strengths:**

- A novel method that integrates insights from SS-DDPM and DPS, achieving excellent performance in few-step generation under specific scenarios.

- The proposed method significantly reduces the requirement for the number of timesteps in the backward process, which is a notable advantage for practical applications.

- This paper is well-structured and logically coherent, with smooth transitions between sections. The examples are appropriately designed, making the core ideas easy to follow.

**Weaknesses:**

- The definition of "end-to-end" in the context of FLDD remains unclear. It is necessary to clarify whether this refers to training without separate stages or other specific characteristics.

- In Section 4.2, FLDD exhibits poor performance on the ROCStories dataset, but the authors' explanation for this phenomenon lacks sufficient persuasiveness and requires further elaboration.

**Questions:**

1. In the forward process, what are the key differences between FLDD and Star-shaped Diffusion? Could the authors elaborate on the unique design choices of FLDD in this regard?

2. Compared with continuous diffusion models that adopt similar training strategies on continuous datasets (e.g., MNIST), what specific advantages does FLDD offer? For example, in terms of generation quality, training efficiency, or adaptability to data characteristics.

3. In the experimental results (Table 1), FLDD performs poorly on real-world data, particularly with a PPL of 60, which indicates an inability to generate semantically meaningful sentences. Could the authors provide a more specific analysis of the underlying reasons? For instance, whether it is due to mismatches between the forward process design and real-world data distributions or limitations in the learnable marginal/posterior functions.

4. While FLDD demonstrates strong performance in few-step diffusion, what are the precise time and computational costs compared to existing methods? A smaller number of steps does not always translate to lower overhead, so quantitative comparisons would help verify its efficiency advantages.

---

> ### Author Response · Authors · 2025-11-19
>
> We thank the reviewer for their thoughtful and positive assessment of our work and for highlighting its novelty, clarity, and practical relevance. Below, we address the questions and comments raised in the review.
>
> **Weaknesses:**
>
> 1. We apologise for any confusion regarding terminology. By *end-to-end* we mean that both the forward and reverse processes are trained jointly from scratch under a single ELBO objective, without any teacher model or distillation stage. We optimize the standard diffusion ELBO, where parameters of the reverse and the forward appear together, and gradients for both sets of parameters are propagated jointly. We therefore do not require separate or alternating training stages. At test time, we use exactly the same few-step sampler that was trained with this objective.
>
>     We will state this explicitly in the camera-ready version, if accepted, to avoid ambiguity.
>
> 2. Please see the response to Question 3 below.

---

> ### Author Response · Authors · 2025-11-19
>
> **Questions:**
>
> 1. The main idea of Star-Shaped Diffusion [1] is that its forward process generates noisy samples at intermediate time steps completely independently (so its graphical model looks like a star). This implies, firstly, that $q(z_s|z_t, x) = q(z_s|x)$, and secondly, that from a theoretical perspective (and to obtain good sample quality in practice) the reverse (generative) process must also be non-Markovian.
>
>     In FLDD, while the forward process is also non-Markovian, its latents at different time steps are much more strongly correlated. As a result, its generative process remains Markovian, as in conventional diffusion models. The formulation of FLDD’s forward process is more closely related to DDIM [2], where the forward dynamics can be seen as Markovian when conditioned on $x$. From this perspective, SS-DDPM can be viewed as an extreme case of a DDIM-like formulation, and FLDD can be seen as a generalisation of SS-DDPM to categorical data.
>
>     The main difference between these two approaches is, of course, that SS-DDPM keeps the forward process fixed, while FLDD learns the noising process from data. Moreover, while the substitution of marginals from categorical SS-DDPM into FLDD is straightforward, it is not clear how to efficiently use the FLDD parameterisation of the forward marginals within the SS-DDPM framework.
>
>     Therefore, while SS-DDPM is related to our work, we believe it is still quite different. We will discuss the connections to SS-DDPM in more detail in the camera-ready version, if accepted.
>
> 2. The main distinguishing feature of FLDD is that it allows training a few-step generative process for discrete data. To train a few-step generator, continuous diffusion-based models usually rely on existing continuous dynamics (in continuous space) and differentiable mappings, none of which exist in the discrete case. Moreover, continuous few-step generators typically use some form of distillation for training, which significantly complicates training, increases its cost, and is known to behave unstably. All of this makes it especially difficult to learn a few-step generative procedure for discrete data, and this is precisely the challenge that FLDD addresses.
> 3. We would like to note that on the ROCStories dataset, FLDD demonstrates comparable performance to other fully discrete diffusion models such as SEDD [3]. Some works, such as [4, 5], indeed report better performance for large numbers of steps ($T>100$). However, these methods are specifically designed and heavily tuned for text generation. They use various additional techniques, such as self-conditioning, classifier-free guidance, training with extra losses, and extensive hyperparameter tuning, to improve text sample quality. Moreover, they run diffusion not in the original discrete space but in a latent space, and they rely on powerful encoder–decoder architectures. In our work, we do not use any of these techniques.
>
>     We present FLDD as a general framework for learning a few-step discrete generative process. The main point of our experiments is to demonstrate that FLDD achieves generative performance comparable to baseline methods under a similar computational budget, while losing only a small amount of performance when the number of steps is significantly reduced (regimes in which other models fail to produce any meaningful samples at all).
>
>     We believe that when combined with additional techniques that are known to work well for specific practical tasks, FLDD can achieve even better results. We leave such task-specific extensions for future research.
>
> 4. A single training iteration of FLDD takes approximately twice as long as that of conventional diffusion models that do not parameterise the forward process. A higher computational budget per training iteration is an inevitable consequence of a more flexible forward process. However, we would like to emphasise that the exact factor is a consequence of our design choices. For simplicity of the experimental setup, we use neural networks of the same size to parameterise both the forward and reverse processes. We believe that the parameterisation of the forward process can be made significantly lighter than that of the reverse one, but we leave this hyperparameter optimisation for future work.
>
>     We also note that the forward process is not involved in the generative procedure at all. Thus, while FLDD reduces the number of steps, it does not introduce any additional computational overhead at sampling time.
>
>
> We hope that these clarifications, which further highlight the novelty and practical relevance of our approach, will strengthen your support for the acceptance of our paper!
>
> [1] Okhotin, Andrey, et al. "Star-shaped denoising diffusion probabilistic models."
>
> [2] Song, Jiaming, et al. "Denoising diffusion implicit models."
>
> [3] Lou, Aaron, et al. “Discrete diffusion modeling by estimating the ratios of the data distribution”

---

### Meta-Review · Area_Chair_jCR1 · 2026-01-08

**Summary:**

The authors propose a discrete diffusion framework, Forward-Learned Discrete Diffusion (FLDD), improves sampling efficiency. The primary contribution is in a non-Markovian forward process with learnable marginal and posterior distributions, which allows the generative process to remain factorized while matching the target distribution defined by the noising process. The method is demonstrated across synthetic data, binarized MNIST, text generation, and molecule-generation tasks. Modest degradations were observed when the number of diffusion steps goes down from 100 to 10 for molecular generation.

The initial recommendations for this manuscript were 2 marginally above acceptance threshold (mvjP, vLZq), 1 marginally below the acceptance threshold (ATG6), and 1 accept (wcR4). While several reviewers commented that the work is novel, there were a number of questions and concerns raised: (1) The definition of "end-to-end" in the context of FLDD remains unclear (mvjP); (2) poor performance on the ROCStories dataset (Table 1), but the authors' explanation for this phenomenon lacks sufficient persuasiveness and requires further elaboration (mvjP, ATG6); (3) details on key differences between FLDD and Star-shaped Diffusion (mvjP); (4) lack of details on specific advantages that FLDD offer (mvjP); (5) lack of details in time and computational costs compared to existing methods and how they affect performance (mvjP, vLZq); (6) overly simplistic experiments on 2D Gaussian mixture and binarized MNIST (ATG6); (7) lack of details on computational costs (wcR4); (8) does not compare against other standard gradient estimators nor discuss control variates (wcR4).

The AC notes that review of vLZq did not touch on methodological aspects.

Overall, the authors were able to address most of the questions and concerns raised by the reviewers. One point of contention was whether the empirical results presented were compelling. While the author claims that they are comparable to existing methods, this seems to only hold true for molecular and not ext generation. Hence, the AC advises the authors to tone down their statement. Nonetheless, the AC sees merit in the work. The AC recommends the authors to consider the outstanding points and incorporate the feedback and materials presented in the rebuttal, which would improve the next revision of the manuscript.

**Reviewer Concerns:**

The authors posted a rebuttal to address the following points:

(1) The definition of "end-to-end" in the context of FLDD remains unclear (mvjP):

The authors elaborated that the term end-to-end refers to training from scratch using ELBO objective for both forward and backward processes. This point has been addressed.

(2) poor performance on the ROCStories dataset (Table 1), but the authors' explanation for this phenomenon lacks sufficient persuasiveness and requires further elaboration (mvjP, ATG6):

The authors noted that other works reported better results because they have been specifically tuned for text generation. The authors sees FLDD as a general framework without the need for specific tuning, architecture, and training techniques. While the authors mentioned that FLDD can achieve even better results with these techniques, they do not provide evidence.

Additionally, the authors argue that FLDD achieves generative performance comparable to baseline methods under a similar computational budget. The AC checked Tables 1, 2 and agree with the reviewers that there is a gap between the proposed method and existing methods even for T=100 in Table 1. In Table 2, the proposed method (T=10, T=100) is comparable to existing methods, but it also seems that entire benchmark is saturated with little difference across methods. This point is was not addressed.

(3) details on key differences between FLDD and Star-shaped Diffusion (mvjP):

The authors discussed the connection and difference to star-shaped DDPM. This point has been addressed.

(4) lack of details on specific advantages that FLDD offer (mvjP):

The authors clarified that the main advantage offered is few-step generative process for discrete data. This point is addressed.

(5) lack of details in time and computational costs compared to existing methods and how they affect performance (mvjP, vLZq):

The author stated that a single training iteration takes twice as long as that of convention diffusion models. The authors mentioned that the forward process can be made made lighter, but the forward process is not involved in the generative procedure.

(6) overly simplistic experiments on 2D Gaussian mixture and binarized MNIST (ATG6):

The authors agree with the assessment that 2D Gaussian mixture and binarized MNIST alone are not strong enough of evidence for the proposed method; the main evidence is in text and molecular generation. This point has been addressed.

(7) lack of details on computational costs (wcR4):

The authors did not report quality vs total training and inference budget, but does mention that FLDD takes approximately twice as long as that of conventional diffusion. This point was partially addressed.

(8) does not compare against other standard gradient estimators nor discuss control variates (wcR4):

The authors mentioned that they did experiment with Gumbel, Rao–Blackwellization, leave-one-out, REBAR, and RELAX. This point has been addressed.

**Reviewer Scores:**

The AC has read the reviews and the rebuttal. Given the responses, the AC believes all will likely maintain their scores as most of their points were addressed.

---

### Decision · Program_Chairs · 2026-01-26

Accept (Poster)